# On Modelability and Generalizability: Are Machine Learning Models for Drug Synergy Exploiting Artefacts and Biases in Available Data?

## Abstract

Synergy models are useful tools for exploring drug combinatorial search space and identifying promising sub-spaces for in vitro/vivo experiments. Here, we report that distributional biases in the training-validation-test sets used for predictive modeling of drug synergy can explain much of the variability observed in model performances (up to $0.22\ \Delta AUPRC$). We built 145 classification models spanning 4,577 unique drugs and 75,276 pair-wise drug combinations extracted from Drug-Comb, and examined spurious correlations in both the input feature and output label spaces. We posit that some synergy datasets are easier to model than others due to factors such as synergy spread, class separation, chemical structural diversity, physicochemical diversity, combinatorial tests per drug, and combinatorial label entropy. We simulate distribution shifts for these dataset attributes and report that the drug-wise homogeneity of combinatorial labels most influences modelability ($0.16 \pm 0.06\ \Delta AUPRC$). Our findings imply that seemingly high-performing drug synergy models may not generalize well to broader medicinal space. We caution that the synergy modeling community's efforts may be better expended in examining data-specific artefacts and biases rigorously prior to model building.

## 1 Introduction

For complex, multifactorial diseases such as cancer, combination therapies offer the possibility of enhanced efficacies [19], with reduced effective doses and associated host toxicities [9], as well as a strategy for slowing the evolved drug resistance commonly observed in monotherapies [32]. It is, however, more challenging to perform clinical trials for combination therapies [22] and the large number of possible drug combinations renders exhaustive testing by brute-force heuristics infeasible. Machine learning is a useful tool for exploring the vast drug combinatorial search space and identifying promising sub-spaces for in vitro/vivo experiments.

Currently, research in the field of predictive modeling for drug synergy is largely focused on model generation and the optimization of performance metrics such as AUC (which overestimates model performance on imblanaced datasets [30, 15]), rather than the context in which models are generated and deployed. Model improvements are not reported in tandem with descriptive statistics characterizing the quality and modelability of datasets. Nair et al. [18] proffer that a limitation of their dataset is that drug combination screens are generally discordant across independent studies. There is no consensus definition for drug synergy [17, 29] and the experimental endpoints modeled are often proxies of drug response that can be easily measured in a high-throughput fashion but lack clinical relevance or even reproducibility [20].

Biases have been reported in datasets used for model generation in adjacent research fields, such as PDBBind and CASF for the prediction of ligand-protein binding affinities [27]. In a systematic review of 41 genomic machine learning studies, Barnett et al. [2] investigated which components of a study contributed to improvements in model performance and whether reported improvements represent a true improvement or an unaddressed bias inflating performance. They found that data leakage due to feature selection and the number of hyperparameter optimizations were significantly associated with an increase in reported model performance. In a review of 62 machine learning studies on the detection and prognostication of COVID-19 using chest radiographs and chest computed tomography images, Roberts et al. [26] found that none of the models identified were of potential clinical use due to biases in either the methodology or underlying data.

Previous studies on drug synergy prediction have not examined artefacts and biases in dataset composition. To the best of our knowledge, no attempt has been made to quantify the sensitivity of synergy models to underlying distributions in either input feature or output label spaces. Alsherbiny et al. [1] note that the source of drug combination screening data, i.e. NCI-ALMANAC [8] versus ONEIL [21], has a more significant impact on model performance than feature engineering. Similarly, Rani et al. [25] note that synergy models built using NCI-ALMANAC tend to outperform those built using ONEIL. Here, we report that distributional biases in the datasets used for predictive modeling of drug synergy explain much of the variability observed in model performances (up to $0.22 \ \Delta AUPRC$). We built 145 binary classification models using drug combination screens extracted from DrugComb [35] spanning 4,577 unique drugs and 75,276 pair-wise drug combinations. We characterize the central tendencies and dispersions of various dataset attributes, and subsequently simulate distribution shifts to demonstrate that model performance can improve or deteriorate depending on the direction of attribute shift.

## 2  Methodology

### 2.1  Synergy Definition

We use the Bliss Independence model [3], one of several synergy reference models [17, 29], to qualify and quantify the expected additive or null response of administering a drug combination. Operating under assumptions of statistical independence between drugs (i.e., the modes of action of constituent drugs in a combination differ), symmetry in drug interactions, no variability in responses, and continuous dose-response relationships, Bliss excess is defined mathematically as:

$$E_{Bliss} = E_{AB} - (E_A + E_B - E_A \times E_B)$$

where $E_{AB}$ is the observed effect of the drug combination, and $E_A$ and $E_B$ are the observed individual effects of drugs A and B, respectively. $E_{Bliss} = 0$ is the threshold for additivity, while $E_{Bliss} > 0$ indicates synergy and $E_{Bliss} < 0$ indicates antagonism.

### 2.2  Data Collection and Pre-Processing

Drug pair synergy data targeting 142 cancer cell lines and 3 malarial parasites was extracted from DrugComb v1.5 [35]. Thirty-three percent of drug-drug-cell line tuples were replicate experiments, which we deduplicated by computing the geometric mean synergy score across replicate samples. Thirty-nine percent (N = 306,282) of the combination-cell line tuples were sourced from NCI-ALMANAC [8] and twenty-five percent (N = 198,722) were sourced from FRIEDMAN [12], with the remainder sourced from twenty-two other combination screens including ONEIL [21] (twelve percent; N = 92,208) and CLOUD [14] (five percent; N = 40,160). In total, 75,276 pair-wise drug combinations comprising 4,577 unique drugs were obtained for 145 cell-line synergy endpoints defined by the Bliss Independence model. We selected the top and bottom fifteen percent of each cell-line dataset's distribution of Bliss synergy scores to obtain balanced classes after filtering out additive samples.

## 2.3 Dataset Attributes and Metrics

**Synergicity**   Synergicity measures the degree to which a given drug is associated with synergistic combinatorial labels: it is defined in this work, as in previous work [34], as the fraction of combinations for which individual drugs have been labelled synergistic as opposed to antagonistic. At the cell-line dataset level, the interquartile range or H-spread was used to capture the bimodality of synergicity distributions and test the hypothesis that cell-line datasets with drugs found primarily in antagonistic-only combinations (`synergicity = 0`) and synergistic-only combinations (`synergicity = 1`) are easier to model with higher AUPRC scores.

**Combinatorial Label Entropy**   Combinatorial label entropy measures the level of disorder or heterogeneity of combinatorial labels. It is defined mathematically as Shannon entropy:

$$H(X) = -\sum_{i=1}^{n} P(x_i) \log_2(P(x_i))$$

where $H(X)$ is the Shannon entropy of a discrete random variable $X$ and $P(x_i)$ is the probability of outcome $x_i$ occurring in the system. The sum is taken over all $n$ possible outcomes $x_i$. In our case, $H(X)$ has range [0, 1] and measures how homogeneous the combinatorial labels associated with a given drug are: if a drug occurs predominantly in drug combinations labelled synergistic-only or antagonistic-only, then its combinatorial label entropy is low (close to 0); if a drug occurs in drug combinations labelled synergistic approximately half of the time and antagonistic approximately half of the time, then its combinatorial label entropy is high (close to 1).

**Feature Similarity**   Feature similarity in chemical structural and physicochemical spaces was defined in two steps: cosine similarity computed pair-wise amongst all drugs tested per cell line, followed by the cell-line fraction of pair-wise similarities above 0.15. Mathematically, the cosine similarity between two feature vectors $A$ and $B$ is defined as:

$$\text{cosine\_similarity}(A, B) = \frac{A \cdot B}{\|A\| \cdot \|B\|}$$

**Non-Additivity**   A drug's tendency for non-additivity when combined was scored as the median absolute distance from Bliss additivity across combinations. This measure was used to test the hypothesis that a drug's combinatorial label entropy decreases with its tendency for non-additivity in combinations. In other words, non-additivity thus defined was used to test whether the degree of synergism or antagonism achieved by a drug was associated with the consistency or homogeneity of its combinatorial labels.

## 2.4 Model Generation and Evaluation

We formulate drug synergy prediction as a supervised classification task: we construct one binary model per cell-line dataset, resulting in a total of 145 binary models, to predict synergistic versus antagonistic class labels for drug-drug pairs using the CRAN "randomForest" [13, 24] implementation of the traditional random forest learner by Breiman [4] under default hyperparameter optimizations. Given that the focus of this work is the influence of dataset composition on model performance, and not the influence of model architecture on model performance, we required a single learner to serve as our baseline before and after shifting attribute distributions. We deliberately chose a decision tree ensemble learner as our baseline due to its computational efficiency on high-dimensional data, adequate interpretability and explainability, as well as state-of-the-art model performance on balanced and minority classes [6]. We constructed two sets of drug features: structural 2048-bit Morgan fingerprints (with radius 3) and 43-element long physicochemical profiles of all available molecular descriptors on RDKit [11]. Feature vectors were concatenated for each drug-drug pair in both permutations. Our 80%-20% train-test split strategy was drug-pair–stratified with five-fold cross-validation. To evaluate model performance, we computed Area under the Precision-Recall curve (AUPRC), which is less sensitive to class imbalance and thus more practically relevant and

actionable than Area under the Receiver Operating Characteristic curve (AUROC) [30, 15]. The mean AUPRC across all models ($n = 145$) was 0.76 ± 0.09. For our categorical analyses, we categorized cell-line models with AUPRC greater than or equal to 0.8 as high-performing ($n = 50$), and cell-line models with AUPRC less than 0.8 as low-performing ($n = 95$).

## 2.5 Simulating Distribution Shifts in Dataset Attributes

We simulated distribution shifts in dataset attributes by sub-sampling each cell-line dataset. For originally high-performing models, we selected subsets of drugs with high combinatorial label entropy (upper 15%), few combinatorial tests per drug (lower 15%), low physicochemical similarity to other drugs (lower 15%), and low structural similarity to other drugs (lower 15%). Conversely, for originally low-performing models, we selected subsets of drugs with low combinatorial label entropy (lower 15%), many combinatorial tests per drug (upper 15%), high physicochemical similarity to other drugs (upper 15%), and high structural similarity to other drugs (upper 15%). This simulated shifts in attribute distributions such that high-performing models now resembled low-performing models, and vice versa. Cell-line models with insufficient drugs remaining were discarded, yielding 103 models for structural similarity, 109 models for physicochemical similarity, 117 models for combinatorial tests per drug, and 91 models for combinatorial label entropy per drug. The simulations were run for each of the dataset attributes identified individually, as well as pair-wise, but the latter yielded datasets too small for model generation. To distinguish change in model performance due to shifting bias versus reduction in dataset size, models were trained, validated, and tested on shifted and non-shifted subsets of comparable size for each cell line.

# 3 Results

## 3.1 Synergy Spread and Class Separation

We first analyzed the effect of dataset span, measured as standard deviation of Bliss synergy scores, and class separation, measured as difference in mean Bliss synergy scores of antagonistic vs synergistic classes, on cell-line model performance, measured as AUPRC. The results are shown in Figure 1. It can be seen that high-performing cell-line models tended to exhibit broader synergy spread with difference in means between high– and low–performing models of 15.4–24.1 (95% CI) Bliss synergy units (Welch's two-sample t = 9.13, df = 71.3, p = 1.26e-13). This is consistent with the relationship between potency span and achievable model performance reported by Brown et al. [5] in the context of predicting binding affinity of small-molecule ligands for protein targets. High-performing cell-line models also tended to exhibit greater class separation in synergy space with difference in means between high– and low–performing models of 12.9–17.6 (95% CI) Bliss synergy units (Welch's two-sample t = 13.1, df = 94.4, p < 2.20e-16). Easier class splits may inflate model performance, particularly on AUROC [30, 15] but also AUPRC: DeepSynergy, for instance, defined the top 10% of combinations as the synergistic or positive class and modeled the remainder as the negative class [23]. Our findings show that both synergy spread and class separation influence modelability.

## 3.2 Synergicity and Entropy of Combinatorial Labels

We then analyzed the effect of combinatorial label homogeneity on model performance (Sub-Figures 2A-B). It can be seen that the cell-line H-spread of synergicity, defined as the fraction of combinations for which individual drugs have been labelled synergistic as opposed to antagonistic, is positively correlated with cell-line model performance, measured as AUPRC (Spearman's $\rho = 0.539$, p = 1.77e-10). Conversely, the cell-line arithmetic mean heterogeneity of combinatorial labels, measured as Shannon entropy for individual drugs, is negatively correlated with cell-line model performance, measured as AUPRC (Pearson's $r = -0.691$, p < 2.20e-16). The more bimodal a cell line's drug synergicity distribution, the more homogeneous its drug-wise combinatorial labels and the easier to predict combinations unseen during training with at least one seen-before drug. Our findings imply that cell lines comprising drugs with homogeneous combinatorial labels, i.e., drugs occurring

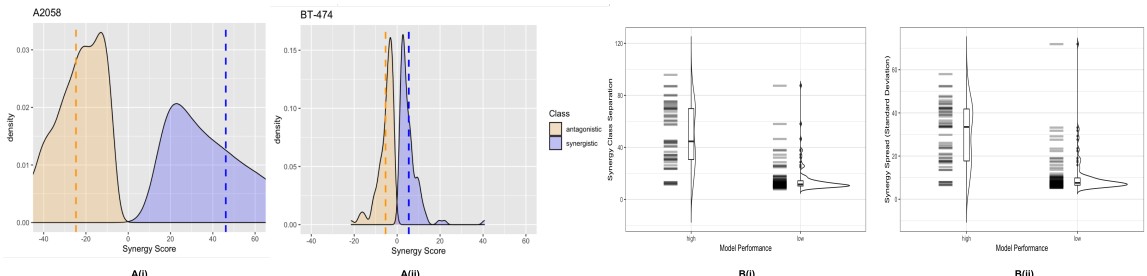

Figure 1: **Panel A.** Distribution of Bliss synergy scores for the best-performing cell-line model, **A(i)**, and the worst-performing cell-line model, **A(ii)**. **Panel B.** Each barcode line in the violin plots represents one cell-line model. Differences in synergy class means, **B(i)**, and standard deviations of overall synergy distributions, **B(ii)**, for all cell-line models binned into high versus low AUPRCs.

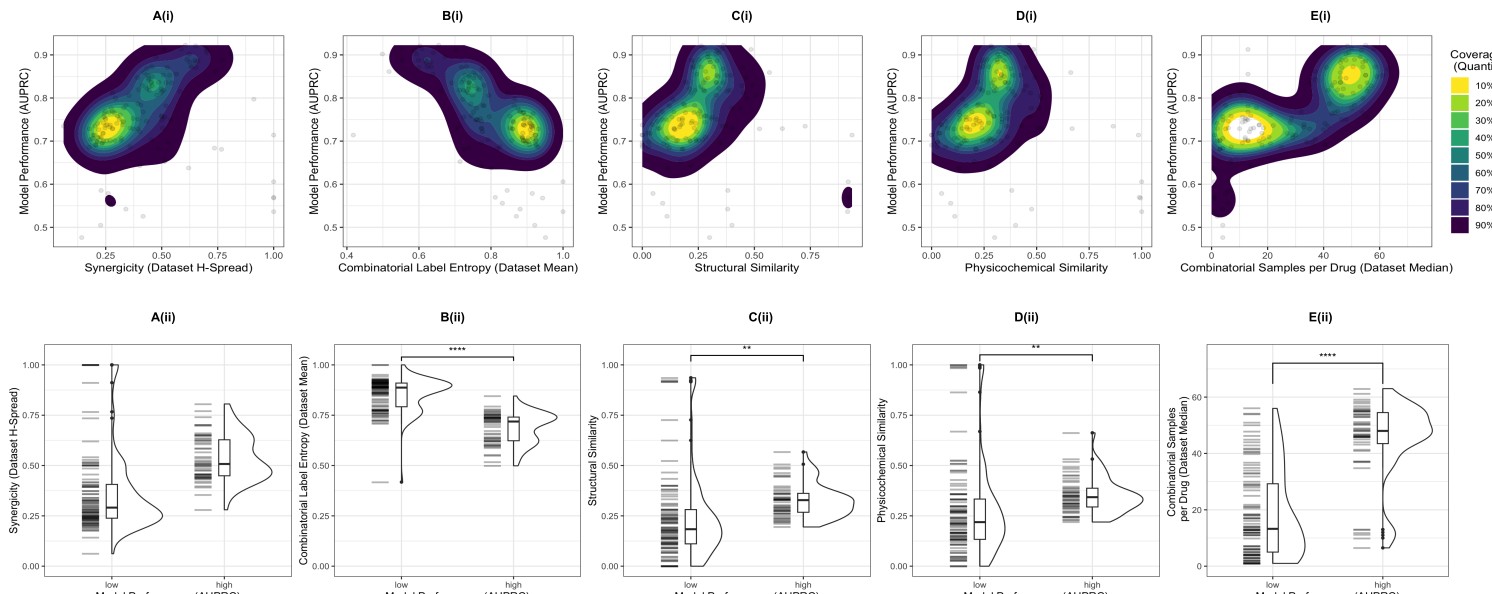

Figure 2: **Panel A.** Density and violin plots of cell-line H-spread of the fraction of combinations for which individual drugs have been labelled synergistic (dubbed synergicity) and cell-line model performance (Spearman's $\rho = 0.539$). **Panel B.** Density and violin plots of cell-line mean combinatorial label entropy and cell-line model performance (Pearson's $r = -0.691$). High-performing cell-line models exhibited lower diversity spanning 3.91%–13.8% (95% *CI*) higher cosine similarity in structural space with Spearman's $\rho = 0.359$ (**Panel C**) and 2.28%–12.9% (95% *CI*) higher cosine similarity in physicochemical space with Spearman's $\rho = 0.327$ (**Panel D**), as well as 17.1–31.0 (95% *CI*) more combinations tested per drug with Pearson's $r = 0.504$ (**Panel E**). Each dot in the density plots (upper panels) and each barcode line in the violin plots (lower panels) represents one cell-line model.

169  primarily in antagonistic-only combinatorial labels and synergistic-only combinatorial labels, tend to
170  be easier to model with higher AUPRC scores.

## 3.3 Structural Diversity, Physicochemical Diversity, Combinatorial Tests Per Drug

172  We then analyzed the effects of drug diversity in structural Morgan fingerprint and physicochemical
173  spaces, both measured as fraction of drugs in a cell-line dataset with pair-wise cosine similarity
174  above a defined threshold, on cell-line model performance, measured as AUPRC. Panel C of Figure 2
175  shows that the dataset attribute, compound structural similarity, is positively correlated with model

performance (Spearman's $\rho = 0.359$, p = 1.012e-05): high-performing cell-line models exhibited 3.91%–13.8% (95% CI) higher pair-wise cosine similarity between drugs in Morgan fingerprint space than low-performing cell-line models (Welch's two-sample t = 3.54, df = 132.64, p = 0.0005). Similarly, Panel D of Figure 2 shows that the dataset attribute, compound physicochemical similarity, is positively correlated with model performance (Spearman's $\rho = 0.327$, p = 6.282e-05): high-performing cell-line models exhibited 2.28%–12.9% (95% CI) higher pair-wise cosine similarity between drugs in physicochemical space than low-performing cell-line models (Welch's two-sample t = 2.83, df = 131.33, p = 0.005). Summarily, the breadth of compound structural and physicochemical spaces both appear to influence modelability, which one might expect as it is easier to model a smaller space with greater overlap between train and validation/test sets. We subsequently investigated the relationship between cell-line model performance, measured as AUPRC, and number of combinatorial tests per drug. It can be seen in Panel E of Figure 2 that this dataset attribute is positively correlated with model performance (Pearson's $r = 0.504$, p = 1.24e-10). High-performing cell-line models comprized 17.1-31.0 (95% CI) more combinations tested per drug than low-performing cell-line models (Welch's two-sample t = 6.86, df = 141.19, p = 1.99e-10), which one might expect as it is easier to model a smaller space with fewer distinct drugs tested in more combinations. These findings imply that seemingly high-performing drug synergy models do not generalize well to broader medicinal space.

### 3.4 Simulating Distribution Shifts in Dataset Attributes

To test whether the differences in model performance observed across cell lines was due to underlying data modelability versus biological variability, we simulated shifts in dataset attribute distributions and compared resulting changes in model performance ($\Delta AUPRC$). We selected subsets of drug-drug samples to shift distributions for low-performing cell-line models to resemble high-performing cell-line models, and vice versa. The simulations were run for each of the dataset attributes identified individually, as well as pair-wise, but the latter yielded datasets too small for model generation. The results are summarized in Figure 3.

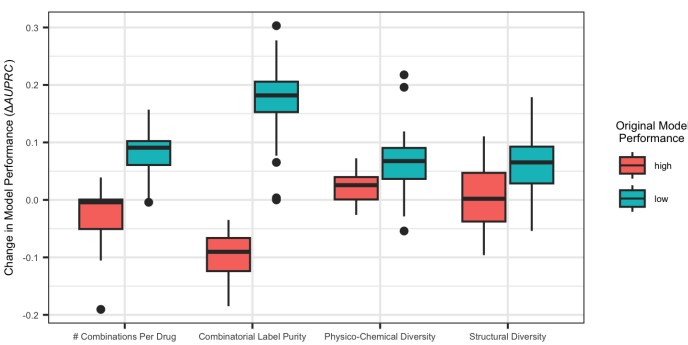

Figure 3: Change in model performance, $\Delta AUPRC$, after simulating distribution shifts for each dataset attribute individually. Attribute distributions for previously low-performing cell-line models were shifted to resemble attribute distributions for high-performing cell-line models, and vice versa. Performance improved for previously low-performing models (blue) under all simulations, albeit to varying degrees ($+0.06 \pm 0.04$ $\Delta AUPRC$ for physicochemical diversity versus $+0.18 \pm 0.05$ $\Delta AUPRC$ for combinatorial label entropy). Performance deteriorated most noticeably for previously high-performing models (red) following shifts in distributions for combinatorial label entropy ($-0.10 \pm 0.04$ $\Delta AUPRC$).

It can be seen that subsetting data points that result in greater class separation, broader synergy spread, lower structural diversity, lower physicochemical diversity, higher number of combinatorial tests per drug, and lower combinatorial label entropy generally increased model performance. Conversely, subsetting data points that result in smaller class separation, narrower synergy spread, lower number of combinatorial tests per drug, and higher combinatorial label entropy generally decreased model

performance. In other words, simulating shifts in attribute distributions tended to boost model performance for originally low-performing models, and tended to degrade model performance for originally high-performing models. This suggests that the differences observed in model performance across cell lines was likely due to differences in dataset composition and not due to inherent biological variation. Of the dataset attributes identified and manipulated, combinatorial label entropy most influenced modelability, increasing the performance of originally low-performing models by $+0.18 \pm 0.05 \, \Delta AUPRC$, which is comparable to the original difference in mean performance between high–versus low–performers ($0.15 \, \Delta AUPRC$). It is important to note that factors are not decoupled in these simulations as shifting one attribute distribution in isolation was not feasible; shifting one distribution simultaneously shifted other distributions to varying degrees since we must also consider how dataset attributes are correlated with each other. To contextualize these findings, we refer to improvements over state-of-the-art models reported in drug synergy literature, such as $+0.04 \, \Delta AUPRC$ by Preuer et al. [23] and Wang et al. [31].

### 3.5 Synergy, Lipophilicity, and Model Performance

We then analyzed whether mechanistic insights reported in drug synergy literature, particularly the relationship between synergicity and lipophilicity [34], influence modelability. Figure 4A shows that, for the well-characterized cell line MCF7, a drug's lipophilicity (CrippenClogP) is positively correlated with its synergicity, measured as the fraction of combinations for which the drug has been experimentally labelled synergistic as opposed to antagonistic, particularly in the region most relevant for drug discovery, i.e., CrippenClogP interval (1,6]. Figure 4B shows the correlation between lipophilicity and synergicity for all cell lines plotted against model performance (Spearman's $\rho = -0.351$, p = 1.575e-05): high-performing models evidently do not rely on the positive correlation between lipophilicity and synergicity reported here and in literature [34] for predictions.

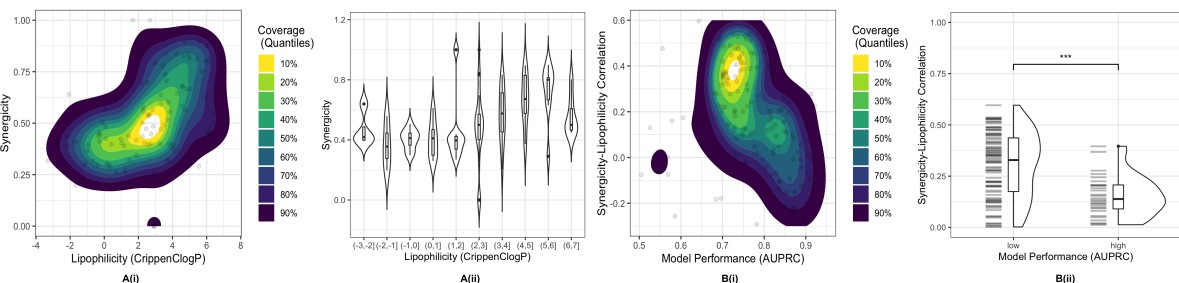

Figure 4: **Panel A.** A drug's lipophilicity (CrippenClogP) is correlated with its synergicity in the MCF7 cell-line dataset, particularly for drug-like molecules in CrippenClogP interval (1,6]. **Panel B.** Correlation between lipophilicity and synergicity plotted as a function of model performance for all cell-line datasets. High-performing models evidently do not rely on the correlation between lipophilicity and synergicity reported here and in literature for predictions.

### 3.6 Non-Additivity, Combinatorial Label Homogeneity, Drug Similarity

We considered the dependence of combinatorial label homogeneity, an output dataset attribute, on various input dataset attributes, such as drug similarity. It can be seen in Appendix Figure 7 that cell-line drug similarity in physicochemical (Pearson's $r = 0.480$) and structural (Pearson's $r = 0.514$) spaces correlate with combinatorial label homogeneity. A drug is more likely to behave generally synergistically or generally antagonistically, or rather elicit mostly synergistic-only or antagonistic-only labels, when combined with similar drugs, since similar drugs hit similar pathways exhibiting homogeneous synergistic *or* antagonistic effect. Different drugs hit different pathways exhibiting heterogeneous synergistic *and* antagonistic effect: synergy with some drugs and antagonism with other drugs depending on pathway hit [16]. We then considered the relationship between a drug's combinatorial label homogeneity and its tendency for non-additivity, defined in this work

as median absolute distance from Bliss additivity across combinations. The correlation between these attributes varied across cell-line models and tended to increase with dataset modelability or increasing model performance in AUPRC (Pearson's $r = 0.378$, Figure 5A). High-performing cell-line models comprized drugs exhibiting a stronger correlation between combinatorial label homogeneity and non-additivity with a 95% *CI* [0.091,0.241] higher Pearson correlation coefficient (PCC) than low-performing cell-line models (Welch's two-sample t = 4.39, df = 114.7, p < 0.00002). $19.4\%$ of cell-line datasets exhibited PCCs between combinatorial label homogeneity and non-additivity $\geq 0.5$. Of these, $75\%$ had model performances AUPRC $\geq 0.8$. Figure 5B shows one such cell-line dataset, namely the skin epithelial-like cell line IST-MEL1, with AUPRC $\geq 0.9$ and PCC between combinatorial label homogeneity and non-additivity $r = 0.643$. In other words, drugs that elicited close-to-additive effects when combined tended to have low combinatorial label homogeneity, while drugs that elicited highly synergistic or highly antagonistic effects when combined tended to have high combinatorial label homogeneity. These findings imply that combinatorial label homogeneity could function as a crude proxy for non-additivity in some contexts, yielding greater modelability.

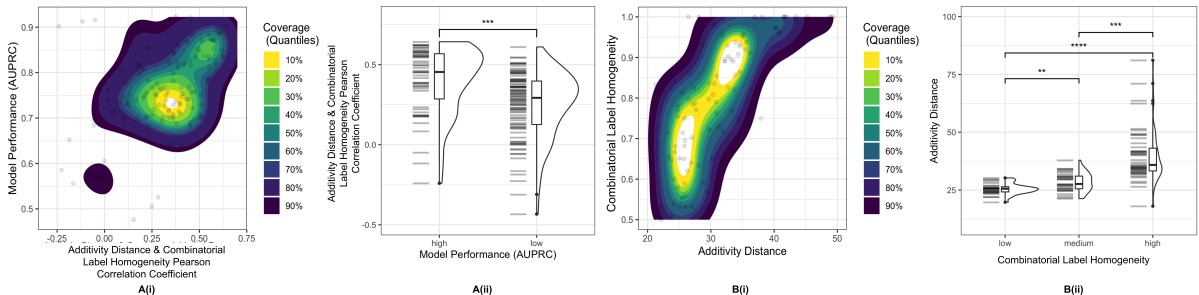

Figure 5: **Panel A.** Each dot in the density plot and each barcode line in the violin plot represents one cell-line model. **Panel A(i).** Model performance (AUPRC) tended to increase with increasing strength of correlation between combinatorial label homogeneity and degree of non-additivity (Pearson's $r = 0.378$). **Panel A(ii).** High-performing cell-line models spanned drugs with a stronger correlation between combinatorial label homogeneity and degree of non-additivity: $95\%$ *CI* [0.091,0.241] difference in mean PCCs. **Panel B.** Combinatorial label homogeneity versus degree of non-additivity for the IST-MEL1 cell line with AUPRC $\geq 0.9$ (Pearson's $r = 0.643$).

## 4 Conclusions

In this work, we qualify and quantify various synergy dataset attributes influencing modelability: synergy spread, class separation, chemical structural diversity, physicochemical diversity, combinatorial tests per drug, and combinatorial label entropy. We simulate shifts in distributions of these attributes and report that combinatorial label entropy improved and degraded model performance most, depending on the direction of attribute shift. It is important to note that the attributes were not decoupled in our simulations as shifting one attribute distribution in isolation was not feasible; shifting one distribution simultaneously shifted other distributions to varying degrees. Overall, our findings imply that model performance is highly sensitive to distributional biases in available data. We find that distributional biases in the training-validation-test sets used for predictive modeling of drug synergy can explain up to $0.22$ $\Delta AUPRC$ of the difference observed in model performances. For comparison, we refer to performance improvements over state-of-the-art models reported in drug synergy literature, such as $0.04$ $\Delta AUPRC$ by Preuer et al. [23] and Wang et al. [31]. We caution that the synergy modeling community's efforts may be better expended in examining data-specific artefacts and biases rigorously prior to model building. We recommend that synergy modelers characterize the applicability domain wherein models can be expected to work reliably and report explicitly the statistical biases underlying datasets used for model generation.

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

# 5 Appendices

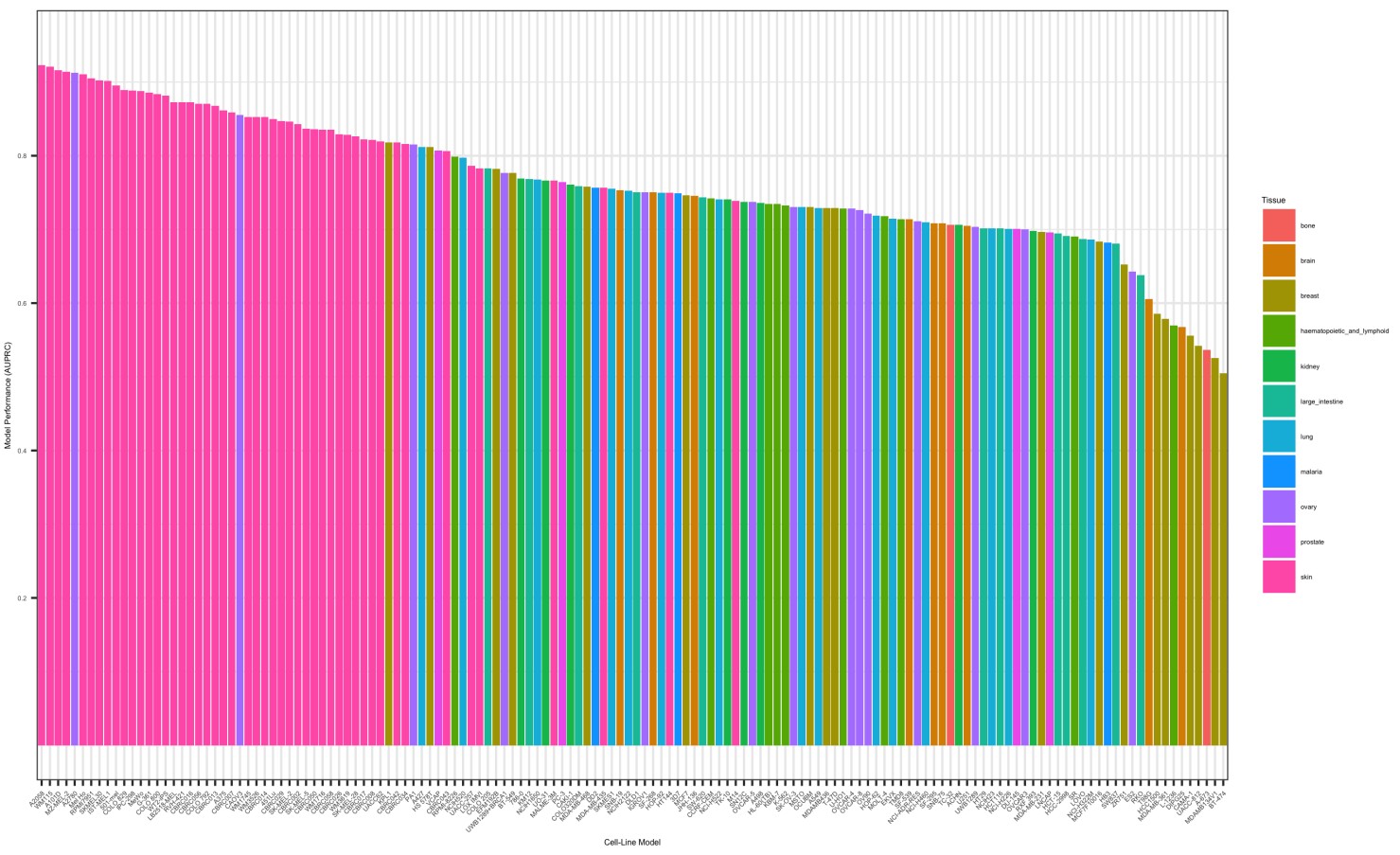

Figure 6: AUPRC performances for all cell-line models investigated in this study.

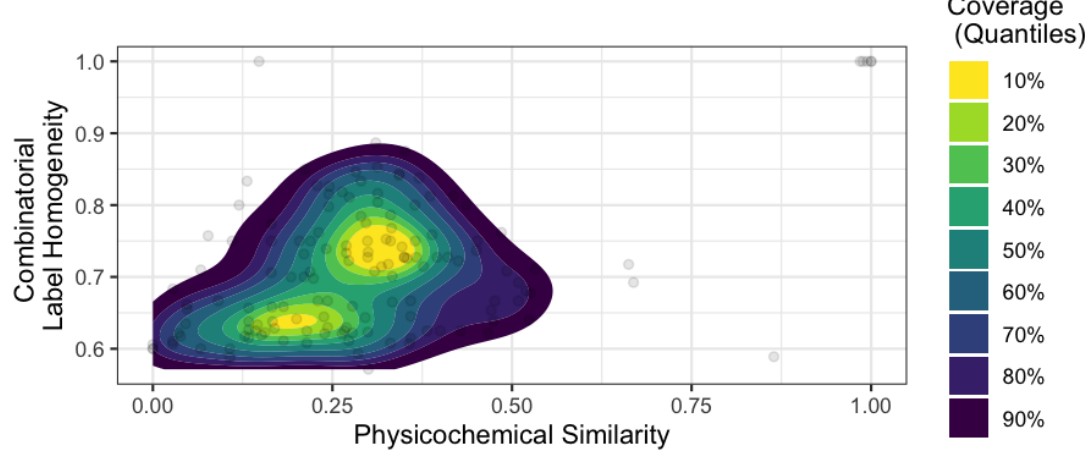

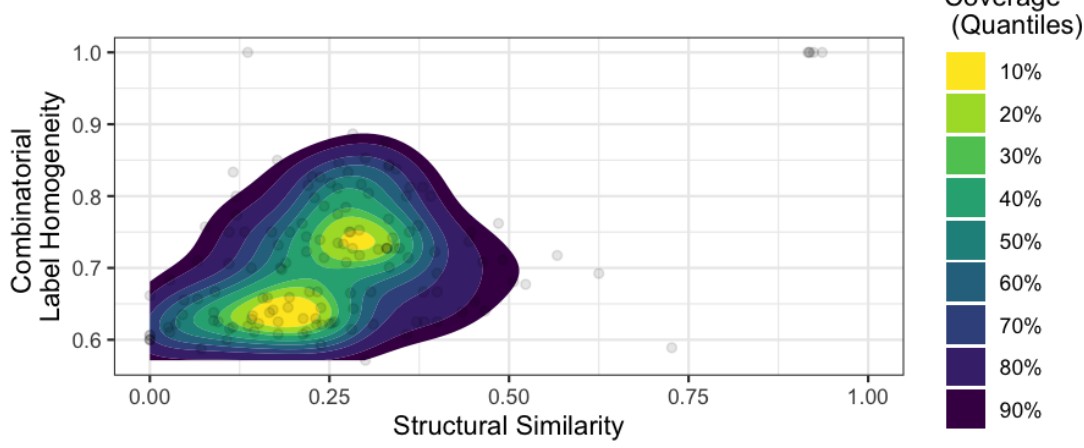

Figure 7: Drug similarity in physicochemical (upper) and structural (lower) spaces correlate with combinatorial label homogeneity.

