# OpenReview forum: "On Modelability and Generalizability: Are Machine Learning Models for Drug Synergy Exploiting Artefacts and Biases in Available Data?"
_NeurIPS.cc/2023/Workshop/AI4Science — NeurIPS2023-AI4Science Poster_

### Official Review · Reviewer_LbXv · 2023-10-23
**Insightful exploration of data biases in machine learning for drug synergy**

**Rating:** 7
**Confidence:** 3

**Review:**

The paper addresses the impact of data biases on the performance of machine learning models for drug synergy. Through a comprehensive analysis, the authors built 145 classification models, encompassing 4,577 unique drugs and 75,276 pairwise drug combinations. They highlight the variability observed in model performances due to distributional biases in the training-validation-test sets. By simulating distribution shifts in dataset attributes, they identify several key factors, such as synergy spread, class separation, and combinatorial label entropy, which have significant effects on model outcomes. The results emphasize that certain dataset attributes can significantly boost or degrade model performance. However, the paper could benefit from clearer delineations between intertwined dataset attributes and more actionable guidelines for addressing the observed biases. In addition, it would be helpful to mention other possible confounding variables or external factors that might have influenced the results. In conclusion, the authors suggest a need for the drug synergy modeling community to rigorously inspect data-specific artefacts and biases before constructing models and call for clear reporting of statistical biases in datasets to ensure reliable model applicability.

---

### Official Review · Reviewer_q1By · 2023-10-23
**Review of Synergy Data Analysis Paper**

**Rating:** 7
**Confidence:** 4

**Review:**

non-additivty  -> non-additivity

The authors should address formatting issue in Figures 1 and 2. Figure 2 is probably too wide. Both captions have issues.
-- I would advise to go through and fix captions in general. It would like nicer without the reverse indentation.

"Different drugs hit different pathways exhibiting heterogeneous synergistic and antagonistic effect: synergy with some drugs and antagonism with other drugs depending on pathway hit [16]." -- feels like it doesn't flow well from the previous sentence. It's a bit unclear in the context of the previous sentence. Can you give an example or reword?

**Summary:**
Overall, I think this paper would be a nice addition to the workshop. The conclusions found in the work generally are quite intuitive. However, the growing performance and popularity of deep models for synergy prediction in the past few years makes this work quite topical since those models generally lack interpretability and are hard to compare. Papers generally seems to use their own subset of synergy datasets to match the features they want to input to their model.

Additionally, multiple recent works (e.g., CancerGPT, SynerGPT) show strong performance with different language models and with surprising inputs to the model. Other work like BLIAM find significant improvement by growing dataset sizes from literature mining. This work may help contextualize the performance of these models. I would have liked to see the authors suggest improvements to datasets for evaluating these models.

It seems that the initial dataset's attributes are the biggest indication of performance. As long as all models are trained and evaluated on the same dataset and splits, do we need to change how benchmarking is done? What features actually matter for predicting drug synergies? Are any deep models actually working? How do these attribute shifts compare to things such as scaffold splitting or zero-shot/few-shot synergy prediction? For future work, I think designing a "balanced" benchmark could be a great contribution. I'm also curious what would have happened using a deep baseline, such as the well-known and simple DeepSynergy. Would the discovered correlations still exist?